# Investigating the Effects of a Multinutrient Supplement on Cognition, Mood and Biochemical Markers in Middle-Aged Adults with ‘Optimal’ and ‘Sub-Optimal’ Diets: A Randomized Double Blind Placebo Controlled Trial

**DOI:** 10.3390/nu14235079

**Published:** 2022-11-29

**Authors:** Lauren M. Young, Sarah Gauci, Lizanne Arnoldy, Laura Martin, Naomi Perry, David J. White, Denny Meyer, Annie-Claude Lassemillante, Edward Ogden, Beata Silber, Andrew Scholey, Andrew Pipingas

**Affiliations:** 1Centre for Human Psychopharmacology, Swinburne University, Melbourne, VIC 3122, Australia; 2Food & Mood Centre, The Institute for Mental and Physical Health and Clinical Translation (IMPACT), School of Medicine, Deakin University, Geelong, VIC 3220, Australia; 3Department of Health Sciences and Biostatistics, Swinburne University, Melbourne, VIC 3122, Australia; 4Centre for Mental Health, Swinburne University, Melbourne, VIC 3122, Australia; 5Department of Nursing and Allied Health, Faculty of Health, Arts and Design, Swinburne University, Melbourne, VIC 3122, Australia; 6Swisse Wellness Pty Ltd., Melbourne, VIC 3066, Australia; 7Department of Nutrition, Dietetics and Food, Monash University, Notting Hill, VIC 3168, Australia

**Keywords:** B vitamins, Bacopa monniera, Ginkgo biloba, cognition, mood, diet quality

## Abstract

***Background:*** Previous randomized controlled trials examining cognitive and mood effects of combination multivitamin supplements in healthy, non-clinical adults have reported mixed results. One purported explanation for this is that the dietary status of participants at the start of supplement interventions may influence the magnitude of the effect of supplementation. ***Methods:*** In this study, we evaluated the effect of a multinutrient formula containing B group vitamins, *Bacopa monniera* and *Ginkgo biloba* on memory, attention, mood and biochemical markers of nutrient status in middle-aged adults (*M =* 52.84 years, *n =* 141) with ‘optimal’ and ‘sub-optimal’ diets over 12 weeks. We hypothesised that active supplementation would differentially improve memory and attention in those with a ‘sub-optimal’ diet. ***Results:*** Mixed model, repeated measures analysis revealed that, in comparison to placebo, active treatment was associated with significant increases in B vitamin status (B1, B6, B12). Regarding behavioural outcomes there was no significant benefit to memory (*F*(1, 113.51) = 0.53, *p* = 0.470) nor attention (*F*(1,113.77) = 1.89, *p* = 0.171) in the whole cohort. Contrary to our hypothesis, there was a significant beneficial effect of supplementation on attentional performance in individuals with an ‘optimal’ diet prior to supplementation (*F*(1,57.25) = 4.94, *p* = 0.030). In the absence of a main effect of supplementation across the entire cohort, there were also a number of significant three-way interactions (treatment by time by diet group) detected in secondary outcomes including lower state anxiety and mental fatigue in those with an ‘optimal’ diet. ***Conclusion:*** These findings suggest that the cognitive benefit of B vitamin and herbal supplementation may be dependent on diet quality, supporting the concepts of ‘co-nutrient optimisation’ and interdependency of nutrients. This warrants further investigation. This study advocates characterising the diet of participants prior to supplementation as it may influence the effect of a nutraceutical intervention.

## 1. Introduction

Modern ‘Western’ diets are insufficient in essential nutrients (Vitamin B, C, D & E) which may lead to poorer long-term health outcomes [1,2,3,4]. Healthy brain function is reliant on these nutrients for the maintenance of healthy function. This has led to growing concern for the increasing proportion of the population who are at risk for poorer cognitive function and mood [5,6,7]. A sub-optimal diet has been implicated as a potential risk factor for dementia [8]. In the face of this, a significant number of individuals from Western countries have reported the use of nutraceutical supplements [9,10,11]. These formulations not only offer opportunity to compensate for nutritional inadequacies but there is growing evidence that they may also optimise psychological functioning into older age.

While it is widely accepted that vitamin supplementation can redress symptoms associated with frank nutrient deficiency, the effect in healthy cohorts is more controversial. Nevertheless, there is increasing support from randomized controlled trials of combination multinutrient supplements (especially those which include B vitamins) showing benefits to cognitive function and mood in healthy, non-clinical adults [12,13,14]. For example, Macpherson and colleagues [15] found that supplementation over 16 weeks improved speed of response for a spatial working memory task in elderly (64–82 years) women. Similarly, Harris and colleagues [16] found improvements in contextual recognition memory performance in men aged 50 to 69 years with a sedentary lifestyle following eight weeks of multivitamin supplementation. Both of the tasks showing effects engage short-term memory. In addition to improvements in memory, two trials found improvements in mood following only four weeks of multinutrient supplementation [17,18] A recent meta-analysis further supports the benefit of B vitamins on cognitive function, demonstrating improvements in episodic memory [19]. In healthy adults aged 20–50, multivitamin supplementation over 16 weeks was associated with a trend towards improvement in performance on the incongruent Stroop task, but this effect was only evident in men [20]. In addition, Haskell and colleagues [21] found significant improvements in speed of completing the Stroop task in females aged 25 to 50 years following nine weeks supplementation with a multinutrient containing B vitamins. The Stroop task is a classic test of selective attention which requires engagement of attentional and inhibitory processes, specifically attending to the colour of a font while inhibiting the meaning (the colour which it describes). Taken together, these findings suggest that supplementation may differentially target these aspects of memory and attention.

Despite the potential benefit of multinutrient formulas to support cognitive functioning and mood in healthy, non-clinical adults, a number of studies in such cohorts have reported minimal or null behavioural benefit of supplementation [22,23,24,25]. Interestingly, the majority of these trials had a longer duration (six months to one year), but the supplements under investigation did not contain any herbal constituents such as Bacopa monnieri and Ginkgo biloba [22,23,25]. These plant-derived compounds have been used for health and well-being for hundreds if not thousands of years in Ayurvedic and traditional Chinese medicine [26]. Bacopa monnieri and Ginkgo biloba have been investigated in separate clinical trials demonstrating enhancing effects for cognition [27,28,29,30,31]. Recent systematic reviews found evidence that Ginkgo biloba is associated with improved cognitive function, including perceptual and motor function, while supplementation with Bacopa monnieri has been found to improve language learning and memory [32,33]. For example, Gschwind and colleagues [34] found supplementation with 240 mg extract of Ginkgo biloba (Symfona^®^ forte) in patients with mild cognitive impairment was associated with improved neuro-motor control of gait. However, in a meta-analysis of healthy participants free from cognitive impairment, no positive effects of Ginkgo biloba supplementation was observed [35]. Recently, Lopresti and colleagues [31] found that supplementation with 300 mg of Bacopa monnieri extract daily for 28 days was associated with improvements in emotional well-being, general health, and pain-related symptoms. These studies demonstrate the possible benefits of herbal supplements. The inclusion/exclusion of herbal components in multivitamin supplements may explain the discrepancies between previous trial findings. While herbal constituents may act through cardioprotective pathways [36], B-group vitamins primarily act on neurotransmitter regulation and energy production [37]. The understanding of how these mechanisms may be complementary is still unknown. Moreover, the lack of biomarkers to assess the uptake of herbal constituents is an added challenge for these investigations.

Thus, the current evidence supporting the cognitive efficacy of multinutrient supplements remains inconclusive. One proposed explanation for this is that the methodology used in previous trials may be insensitive to capture the potential effects of supplementation. One methodological aspect is that the age-group of the participant sample under investigation may weaken the potential to detect supplement effects. Some researchers have proposed that studies in older adults (>65 years) may report less efficacious findings as interventions are unable to reverse the damage associated with brain ageing or co-morbidities [38]. The effect of multinutrients in younger adults (<40 years) may not be detectable in this early life stage as such individuals may already be performing optimally. Middle-age presents an at-risk life-stage where the effect of dietary intake and multinutrient interventions on psychological outcomes may be more pronounced [38].

The baseline dietary intake of healthy research participants is also a confounding factor [39,40,41,42]. It has been proposed that those who adhere to a poorer quality diet (characterised by lower intake of essential nutrients) may have a more enhanced response from supplementation due to poorer baseline nutrient status. On the other hand, if participants are already adhering to a nutrient-rich diet it may mask the effect of a nutraceutical intervention [40]. It is currently unclear to what extent that baseline dietary quality may interact with nutraceutical supplementation and whether this may account for the equivocal nature of this research thus far.

The present study aimed to evaluate the effects of a multinutrient formula containing B group vitamins, Bacopa monniera and Ginkgo biloba on memory and attention in healthy, middle-aged adults following 12 weeks supplementation. Measures of subjective mood, stress reactivity and biomarker status were investigated as secondary outcomes. Our primary hypothesis was that there would be significant improvements in both memory and attention after 12 weeks supplementation with a multinutrient formula. A further aim of this study was to evaluate whether baseline diet quality modulated the response to supplementation. Therefore, the study employed a novel recruitment strategy which involved pre-screening participants’ diet quality prior to inclusion in the study [43]. This approach ensured 50% of the cohort adhered to an unhealthy (‘sub-optimal’) diet and 50% adhered to a healthy (‘optimal’) diet. As a secondary hypothesis, it was anticipated that any beneficial effect of multinutrient supplementation would be more pronounced in the ‘sub-optimal’ diet group.

## 2. Materials and Methods

### 2.1. Participants

The trial followed a randomized, placebo-controlled, double-blind, parallel groups design. All participants provided written informed consent before participation. The study was approved by the Swinburne University Human Research Ethics Committee (2017/269) and all procedures were conducted in accordance with International Council for Harmonisation Guideline for Good Clinical Practice (ICH GCP). The trial was registered with the Australian and New Zealand Clinical Trials Registry, and ClinicalTrials.gov, NCT03482063.

Participants were recruited between May 2018 and September 2019 through online advertising and posters around the university. Potential participants were initially screened for eligibility via telephone. To be eligible to participate in the study, participants needed to be between 40–65 years of age, must have been free from neurological conditions, cognitive impairment (Mini Mental State Examination score >24), mood or psychiatric disorders (Beck Depression Inventory score <20; BDI-II), any health conditions that may affect the absorption of food, hypertension (systolic <160 mmHg and diastolic <100 mmHg) and, due to the nature of the cognitive tasks, must have had normal colour vision. Participants were excluded if they were taking medication, herbal extracts, vitamin supplements or illicit drugs that might reasonably be expected to interfere with cognition or mood, or interact with the supplement (as verified by a Medical Practitioner associated with the trial). Female participants could not be pregnant or lactating. The telephone screen included the Australian modified version of the Diet Screening Tool [44,45] which was used to classify participants as either having an ‘optimal’ or ‘sub-optimal’ diet. The study aimed to recruit 50% of participants with an ‘optimal’ diet and 50% with a ‘sub-optimal’ diet. Of the 501 volunteers telephone screened, 160 participants completed an in-person screening visit in the laboratory at Swinburne University where eligibility was confirmed. One hundred and forty-one participants completed baseline assessments and were randomized to study treatment, and 116 participants completed the study (Figure 1). The secondary outcomes included an optional neuroimaging component which a subset of participants completed (*n* = 60). These results will be reported elsewhere.

### 2.2. Dietary Assessment

The Diet Screening Tool (DST) was originally developed by Bailey and colleagues [45] to identify nutritional risk in community-dwelling older adults. The tool has been validated in middle-aged adults, discriminating across nutrient intake and biochemical markers of nutrient status [46]. The Australian modification of the questionnaire [44] was used in the current study. This diet tool consisted of 20 items, 18 items of which related to the frequency of consumption of particular foods and 2 items that related to the number of servings consumed. The DST has a maximum score of 104, with higher scores indicating better diet quality, and less nutritional risk. Higher scores on the DST indicate relatively higher intake of fruit, vegetables, legumes, olive oil and nuts, and a lower intake of processed foods (sweets, chocolate, cakes, processed meats, sugar-sweetened beverages). Participants scoring ≥60 were classified as adhering to an ‘optimal’ diet, while those scoring ≤59 were classified as adhering to a ‘sub-optimal’ diet. These cut-off scores have previously been used to identify nutritional risk, with those scoring ≤59 having significantly lower circulating levels of nutrients including B vitamins [45].

Once enrolled in the study, participants completed a series of Automated Self-Administered 24-h Dietary Assessments (ASA24). Two were completed prior to baseline, one was completed prior to the final visit and one was completed at the final visit. The ASA24 required participants to record the foods, drinks and dietary supplements they consumed over 24 h, even if they did not reflect their usual diet. Participants were asked to not report the study supplement in follow up recalls. Nutrient intakes were calculated from the nutrient composition data from Australian Food, Supplement and Nutrient Database (AUSNUT) 2011-13. The ASA24 provided detailed dietary intake of participants across their participation in the study. While participants were asked to maintain their regular diet and lifestyle behaviours across the trial, baseline macronutrient and micronutrient intake as measured by the ASA24 were compared to follow-up records in order to confirm that any changes in efficacy endpoints could be attributable to the investigational product and not due to other extraneous dietary changes participants may have made over the 12 week supplementation period. 

### 2.3. Cognitive Assessments

Two computerised cognitive batteries; the Swinburne University Computerised Cognitive Aging Battery (SUCCAB) and Cognitive Demand Battery (CDB) were used in this trial to capture the range of cognitive functions that are vulnerable to aging and nutraceutical interventions [15,47,48,49]. The tasks within each battery are described below. To account for the speed/accuracy trade-off that occurs with aging, performance on each task was computed as accuracy (%) divided by speed of response (in milliseconds) [50,51]. The co-primary outcomes were performance on two tasks from the SUCCAB; spatial working memory (memory) and the incongruent trial of the Stroop colour-word task (attention).

#### 2.3.1. Swinburne University Computerised Cognitive Aging Battery (SUCCAB)

For each task, participants were asked to respond as quickly and as accurately as possible using a button box. All tasks were participant-paced, meaning that as soon as the participant responded to a stimulus, the task would automatically present the next stimulus.

*Simple reaction time.* Thirty single white squares were presented in the centre of a computer screen with a randomized inter-stimulus interval (ISI) to avoid anticipation effects. Participants were asked to respond with a right button press as quickly as possible to the appearance of each square. 

*Choice reaction time.* Participants responded to 20 trials with a left (blue) or right (red) button press to the appearance of either a blue triangle or red square, respectively. Presentation order and ISI were randomized to avoid anticipation effects.

*Immediate and Delayed Recognition.* Participants were asked to study a series of 40 abstract images presented in the centre of the screen for three seconds each with no ISI. On completion, a second series of images were presented, half of which were from the studied series and half that were new (Immediate condition). Participants responded with a right (yes) or left (no) button press whether or not they recognized the image from the studied series. This task was repeated at the end of the SUCCAB tasks (Delayed condition). 

*Congruent and Incongruent Stroop.* This test consisted of a congruent and incongruent trial. Stimulus words (RED, BLUE, GREEN, YELLOW) were randomly presented in the centre of the screen in either congruent or incongruent colours. Participants responded by pressing one of four buttons corresponding to the colour of the word, irrespective of what the word read. The incongruent version of this task was used as a measure of selective attention as participants were required to selectively attend to the colour of the word, while inhibiting the automatic reading response.

*Spatial working memory.* In each trial participants were presented with a 4 × 4 white grid on a black background, with six grid positions containing white squares. Participants were given three seconds to remember where the white squares were located. The grid became blank and a series of four white squares were sequentially displayed in various grid positions. Participants responded with a right (yes) or left (no) response to indicate whether each square matched a position that was originally filled. Participants completed 14 trials, each of which were separated by a blank screen displayed for two seconds. This task required participants to hold spatial information in their working memory. 

*Contextual memory.* A series of 20 everyday images were presented at the top/bottom/left/right of the screen for three seconds each with no ISI. On completion of the series, the same images were displayed again in randomized order in the centre of the screen. Participants responded with a top/bottom/left/right button press to indicate the original location of each image. The task required participants to recall the spatial context of the original presentation and was used as a measure of episodic memory.

#### 2.3.2. Cognitive Demand Battery (CDB)

The CDB involves three computerised tasks as described below. Each task was completed sequentially and repeated three times. In total, completion of the battery took 30 min, with the overall cognitive demand increasing over time. The CDB was used to explore cognitive performance under increased cognitive demand. In addition, this task has previously been used in conjunction with Bond Lader scales to measure stress reactivity in response to increased cognitive demand. In this study, we aimed to measure the potential buffering effects of the investigational product for stress reactivity, as measured by the Bond-Lader Visual Analogue Scales, Visual Analogue Mood Scales and State-Trait Anxiety Inventory State Subscale.

*Serial Threes Subtraction.* The participant was required to count backwards in threes from a randomly generated starting number between 800 and 999. The starting number was presented on the computer screen, and then was cleared by the first response. Participants were instructed to respond as quickly and as accurately as possible using the computer keyboard numeric keys. The participant completed the task over a period of two minutes. Total number of correct and incorrect responses were recorded. 

*Serial Sevens Subtraction.* This task was identical to the Serial Threes Subtraction Task, although it involved serial subtraction of sevens. Both the serial threes and serial sevens tasks were used as a measure of working memory, with serial sevens designed to be more difficult than the serial threes subtraction task.

*Rapid Visual Information Processing.* A series of digits were presented in the centre of the screen at the rate of 100 per minute. The participant was required to detect three consecutive ascending odd or three consecutive ascending even digits. The participant responded to the detection of a target string by pressing the ‘space bar’. The task was continuous for five minutes, with eight target strings presented each minute. This task was used as a measure of processing speed. Performance, total number of ‘false alarms’ (responses to non-target strings) and number of ‘missed’ strings (not responding to a target string) were recorded. Consistent with the approach of the SUCCAB cognitive battery described above, performance was calculated by dividing accuracy by response time.

### 2.4. Mood Assessments

While mood outcomes were a secondary outcome of this study, multinutrient supplementation including B vitamins have been shown to provide benefits across a range of mood facets [12,13,17,21]. Therefore, the following measures were chosen as they assess different aspects of mood: including perceived stress, anxiety, depression and overall mood disturbance. These subjective mood questionnaires assessed mood across various time periods (past week to past month). Further, the inclusion of stress reactivity allowed us to capture more acute mood responses in relation to cognitive demand and assess whether supplementation buffered these effects. 

#### 2.4.1. Perceived Stress Scale (PSS) 

The PSS measured the extent to which respondents have perceived events in their life as stressful over the last month. Participants rated their perceived stress on 10 items, each scored on a 5-point scale ranging from ‘never’ to ‘very often’. Higher scores were associated with higher levels of perceived stress.

#### 2.4.2. Profile of Mood States (POMS) 

The POMS required participants to indicate the degree to which they have identified with 65 mood-related adjectives over the past week. Each item is on a 5-point scale from ‘not at all’ to ‘extremely’. A Total Mood Disturbance (TMD) score was computed with higher scores indicative of greater mood disturbance.

#### 2.4.3. The Depression, Anxiety and Stress Scale (DASS)

The DASS comprised of 21 items that pertain to affect-related symptoms for three sub-factors: depression, anxiety and stress. Participants responded to each item to reflect their experiences over the past week. Each item was on a 4-point scale from 0 to 3, with a total range of scores from 0 to 63. Higher scores indicate a higher degree of dysfunction and less desirable affect experience. The DASS is considered relevant for both clinical and non-clinical populations as some experience of such symptoms is considered normal in day to day life.

#### 2.4.4. Stress Reactivity

Before and after both cognitive batteries (SUCCAB and CDB), participants were asked to complete the following subjective experiences of their current mood states. They were asked to respond according to how they felt ‘right now’ in the present moment. The following measures were used to measure subjective experiences of mood both in the absence of cognitive demand (pre-cognitive scores) and in response to cognitive demand (change from pre-cognitive scores). 

*Bond-Lader Visual Analogue Scales (VAS).* The Bond Lader scales have previously been used to measure stress reactivity in response to a cognitive task [52,53]. This task comprised of 16,100 mm lines anchored at either end by antonyms. Participants marked their current subjective state between the antonyms on each line, bearing in mind that each end of the scale represents an extreme. Each line is scored as millimetres to the mark from the negative antonym. From the scores, three factors were determined; ‘alertness’, ‘calmness’ and ‘contentedness’, with higher scores associated with positive mood states. 

*Visual Analogue Mood Scales (VAMS).* Participants were asked to rate their current subjective experiences of stress, anxiety, fatigue, mental concentration and mental stamina on a 100 mm visual analogue scale. For each scale, higher scores indicated higher subjective experience of that current mood state. 

*State-Trait Anxiety Inventory State Subscale (STAI-S).* The State-Trait Anxiety Inventory State Subscale (STAI-S) measured transient experiences of anxiety. Containing 20 items, the questionnaire asked participants to respond with how they are feeling ‘right now’, ranging from ‘not at all’ to ‘very much so’. The total range of scores is from 20 to 80, with higher scores indicating higher levels of anxiety. 

### 2.5. Biochemical Assessment

Fasting venous blood samples were collected by a qualified venepuncture technician or research nurse at the Centre for Human Psychopharmacology, Swinburne University. A serum separator tube (8.5 mL) containing clot activator (silicone and micronized silica) was allowed to clot at room temperature for 30 min before being centrifuged for 10 min at 4000 rpm and then analysed for high sensitivity C-reactive protein, homocysteine and Vitamin B12. A lithium heparin tube (6 mL) was immediately wrapped in foil to prevent degradation of the sample by light and analysed for Vitamin B6 and Vitamin B1. An EDTA tube (4 mL) was also immediately wrapped in aluminium foil and then analysed for Vitamin B2. All samples were pre-processed on site before being sent by courier to a commercial pathology laboratory for analysis. 

### 2.6. Investigational Product

Participants were instructed to take two tablets daily, one tablet during or immediately after breakfast, and one tablet during or immediately after lunch, for the duration of the trial period. The daily dose of the active supplement contained the following; Vitamin B1 (thiamine hydrochloride) 50 mg, Vitamin B2 (riboflavin) 70 mg, Nicotinamide 40 mg, Vitamin B5 (pantothenic acid from calcium pantothenate) 128.26 mg, Vitamin B6 (pyridoxine from pyridoxine hydrochloride) 41.12 mg, Vitamin B12 (cyanocobalamin) 50 µg, Brahmi (Bacopa monnieri) whole plant 7.46 g (equiv. bacosides calculated as bacoside A 46.7 mg), and Ginkgo (Ginkgo biloba) leaf 6 g (equiv. ginkgo flavonglycosides 14.4 mg, ginkgolides & bilobalide 3.6 mg). This formulation is similar to previously cited studies [16,21]. However, Haskell et al. [21] did not include any herbal component. Placebo tablets contained starch and a small amount of riboflavin (2.42 mg) to give them a similar taste and coloration to the active product.

Supplements were dispensed in blister packs with the participant identification number and dispensing date clearly labelled. Participants were requested to abstain from the supplement on the morning of the final visit. Participants were required to return all empty blister packs and unused tablets at the completion of the study. Compliance was calculated as the number of tablets taken, divided the number of days × 2 (to account for the two tablets per day dosing regimen) between their baseline and final visits. A daily log was also used to facilitate compliance, where participants recorded each day when they took their tablets and if any tablets were missed.

Participants were randomly allocated to one of two groups, stratified according to diet status; one receiving active supplementation and the other receiving a placebo formulation. Randomization was completed by an unblinded disinterested third party who was not involved in data collection or analysis and had no interest in nor benefit from the study outcomes. Randomization was generated from randomly sized permutated blocks and stratified according to gender, age and whether or not the participant was participating in the optional neuroimaging component. To enhance the probability of a relatively even allocation of both ‘optimal’ and ‘sub-optimal’ diet subgroups to both active supplementation or placebo groups, two randomization lists were developed; one for individuals with a ‘sub-optimal’ diet, and one for individuals with an ‘optimal’ diet. This ensured that both diet sub-groups had equal chance of being allocated to the active supplementation or placebo group. Randomization codes were kept in a sealed envelope in a locked filing cabinet until analysis of efficacy outcomes was completed. Both study investigators and participants were blind to group allocation.

### 2.7. Procedure

Following successful telephone pre-screening, participants were invited to attend a screening visit at Swinburne University of Technology in Hawthorn, Melbourne. Written informed consent was obtained. Eligibility was confirmed for the following criteria that required in-person screening: absence of cognitive decline using the Mini Mental State Examination (MMSE), absence of mood disturbances using the Beck Depression Inventory (BDI-II) and absence of hypertension using the SphygmoCor device (Model XCEL, AtCor Medical, Sydney, Australia). Anthropometrics, including Body Mass Index (BMI; kg/m^2^), were measured. Participants completed a shortened practice version of the cognitive and mood assessments, so they were familiar with the programs and to reduce practice effects. They then completed their first ASA24 recall with the researcher, which required them to report their entire dietary intake for the previous 24 h. Participants were instructed to complete a second ASA24 recall prior to their next visit at the university, ideally on a weekend, to account for diet intake differences over the week. Following this visit, participants were randomized to receive either the active supplementation or placebo. 

The baseline visit was scheduled within 2 weeks of their screening visit. Prior to this visit, participants were instructed to fast from 10 pm the night before, and to abstain from alcohol and caffeine 12 h before the visit. While fasted, participants had a blood sample taken and were then provided with a standardised breakfast and short break. Participants then completed mood assessments (including stress reactivity), followed by cognitive assessments. Stress reactivity tasks were repeated following the cognitive assessments. At the conclusion of this visit, participants were dispensed their study treatment and a compliance log. The follow-up visit was scheduled 12 weeks ±3 days from the baseline visit. The 12 week visit followed the same procedure as the baseline visit with the following exceptions: upon arrival a count of the remaining tablets was used to calculate compliance and participants indicated whether they believed the study tablets they had been taking contained the active supplementation or placebo. At the 12 week visit they also completed an additional ASA24 recall, identical to the one they completed between their screening and baseline visits. An outline of study assessments included in this paper are displayed in Figure 2.

### 2.8. Statistical Analysis

#### 2.8.1. Sample Size Calculation

The primary outcomes were performance (accuracy % divided by speed of response in milliseconds) on memory (spatial working memory task from SUCCAB) and attention (incongruent version of the Stroop colour-word task from SUCCAB). Previous studies of multinutrient supplementation, with comparable doses of B vitamins, Ginkgo biloba, and Bacopa monnieri, to the current supplement under investigation have found medium sized effects (eta squared = 0.09, Cohens d = 0.63) for short term memory [15] and a trend for significance (eta squared = 0.08) on the incongruent Stroop task in men [20]. Power analysis was conducted using G*Power 3.1.9.2 [54]. Allowing for 80% power, 2.5% significance and an effect of this magnitude (f = 0.29), in a two armed study (active supplementation, placebo) with two assessments (baseline, final), a total sample of 116 was required (alpha = 0.025). To allow for approximately 20% attrition, the study aimed to recruit 140 participants. 

#### 2.8.2. Analysis

Analysis was conducted using IBM SPSS Statistics for Windows version 26 (IBM Corp., Armonk, NY, USA). Intention to Treat (ITT) analysis was conducted including all participants who received at least one dose of the study treatment and satisfied the inclusion/exclusion criteria at enrolment into the study. Baseline characteristics were presented using the mean and standard deviation for continuous variables (median and IQR for non-normal distributions) and percentages for categorical variables. If the assumptions of parametric statistical testing were not met, commonly employed data transformations (e.g., log, square root) were applied. Following any appropriate transformation, Z scores were calculated for each variable and displayed in histogram form. Z scores greater than 3.29 and greatly disconnected from the spread of the data were designated as outliers and excluded from subsequent analysis.

Mixed Model Repeated Measures (MMRM) analyses were conducted using restricted maximum likelihood methods. Treatment by time analysis was fitted with treatment as a fixed between group factor (active supplementation or placebo group) and time as a fixed repeated factor (baseline, 12 weeks). Random intercept models were conducted in the first instance. In the event of the model not meeting convergence criteria, the analysis was modelled with the random intercept removed. Interaction of treatment by time was fitted to the model and any significant interactions were plotted, representing adjusted means after accounting for covariates. As the study was also interested in the influence of dietary status, a treatment by time by diet group was conducted, identical to the aforementioned analysis, with the addition of diet group as a fixed between group factor (‘optimal’, ‘sub-optimal’). Two separate plots (for each diet group) were plotted using the EMEANS function for any significant treatment x time x diet group interactions. Analysis of the two primary outcome measures was conducted at the 2.5% significance level. All analysis beyond this was considered exploratory and therefore conducted at the 5% significance level. 

Macronutrient and micronutrient intake (as derived from the ASA24) at baseline and 12 weeks were compared for each participant using paired samples t-tests. Following unblinding, participants’ guesses as to whether they were taking active supplementation or placebo were compared against their actual allocation using chi-square analysis. The occurrence of at least one adverse event, Adverse Events (AEs), were analysed by treatment group using Fisher Exact tests to determine if there were any significant differences in the occurrence of AEs between active supplementation and placebo. This analysis included all participants, including those who did not complete the study, in order to capture participants who may have withdrawn from the study due to adverse effects of active supplementation. 

#### 2.8.3. Control Variables

The primary control variables used for adjustment were age, gender, and education. Age was measured in years, calculated as the participant’s age at their baseline visit. Gender was coded as male (1) or female (2). Education was measured as years of full-time study. Further, attrition analysis and comparison of treatment groups at baseline were used to identify any other appropriate covariates. Demographic measures and baseline measures for prognostic variables (e.g., blood biomarkers) were compared between participants who completed the study and those who withdrew prior to completing the study using binary logistic regression. Similarly, baseline demographic and prognostic variables were compared across active supplementation and placebo groups using t-tests for continuous variables and chi-square for categorical variables. Any significant differences in baseline measures of demographic information or prognostic variables were identified as covariates.

## 3. Results

### 3.1. Participant Characteristics 

One hundred and forty-one participants completed baseline assessments and were randomized to active supplementation (*n* = 70) or placebo (*n* = 71). While the study aimed to recruit 50% with a ‘sub-optimal’ diet, challenges with recruitment resulted in the cohort comprising slightly more ‘optimal’ diet participants (*n* = 74) than ‘sub-optimal’ (*n* = 67), see [43]. Importantly, allocation to active supplementation and placebo groups remained balanced across diet subgroups. 

Baseline characteristics are presented for all randomized participants in Table 1 (*n* = 141). The cohort was balanced by gender (50.4% female). On average, participants were aged 52.84 years, well-educated (*M* = 16.94, *SD* = 3.36 years of education) and had a high level of physical activity (87% accumulating sufficient amount of activity per week). The average BMI of this cohort was overweight, which is typical for the Australian population. Mean compliance for participants who completed the study was 98% and this did not differ between active supplementation and placebo groups (97.81% vs. 98.34%; *t* = −0.695, *p* = 0.498).

The active supplementation and placebo groups were balanced across demographic variables with the exception of BMI. The active supplementation group had significantly higher BMI than placebo at baseline (28.19 vs. 26.35; *t* = 2.18, *p* = 0.036). In regard to differences in attrition, there was a relatively even dropout rate across active supplementation and placebo groups (*n* = 14 vs. *n* = 11). However, binary logistic regression revealed that individuals who completed the study were more likely to be older, have lower depressive symptoms (BDI-II) and higher blood pressure than those who did not complete the study. As a consequence of these findings, BMI, BDI-II and brachial systolic blood pressure were added into all subsequent analyses, in addition to the pre-defined control variables. 

With respect to diet subgroups, differences in nutrient intake, biochemical markers and psychological outcomes have been reported elsewhere [55]. Briefly, those in the ‘optimal’ diet subgroup were older, had a lower BMI, and significantly higher intake of vitamin E, magnesium, zinc, and fiber (as measured by ASA24) than those in the ‘sub-optimal’ diet subgroup. They also had significantly higher circulating levels of vitamin B6 and red blood cell (RBC) folate. Finally, an inverse, yet not significant relationship was observed between Vitamin B12 and diet screening tool scores. This indicated that the Australian version of the diet screening tool may not be sensitive to capture B vitamin status [55]. 

### 3.2. Primary Analysis

As seen in Table 2, after controlling for the variables identified above (i.e., age, gender, education, BMI, BDI-II, and brachial blood pressure), MMRM analyses revealed no significant treatment by time interaction for attention, as measured by incongruent Stroop performance (*F*(1,113.77) = 1.89, *p* = 0.171), nor for memory, as measured by spatial working memory performance (*F*(1,113.51) = 0.53, *p* = 0.470). There was also no significant 3-way treatment by time by diet group interaction for memory (*F*(1,131.58) = 0.10, *p* = 0.752), suggesting that there was no effect of treatment in either diet subgroup for this task. There was a significant 3-way treatment by time by diet group interaction for attention (*F*(1,113.91) = 4.52, *p* = 0.036). Separate MMRM analyses for each diet subgroup revealed this was driven by a significant treatment by time interaction in the ‘optimal’ diet group (*F*(1,57.25) = 4.94, *p* = 0.030). As displayed in Figure 3, participants with an ‘optimal’ diet who were taking active supplementation had improved attention after 12 weeks active supplementation. Individuals in the sub-optimal group who were taking active supplementation also observed improvements in attention, however this change was comparable to improvements observed in the placebo group. The treatment by time interaction in those with ‘sub-optimal’ diets was therefore not significant (*F*(1,56.55) = 0.372, *p* = 0.544).

### 3.3. Secondary Analysis

#### 3.3.1. Cognition

MMRM analyses revealed no significant treatment by time effects across any other cognitive tasks when studying the entire cohort (see Table 2). There were, however, a number of significant treatment by time by diet group effects, and subsequently, significant treatment by time effects within diet subgroups. Individuals with a ‘sub-optimal’ diet in the active treatment group had significantly lower number of incorrect responses on the serial sevens task following supplementation compared to placebo (*F*(1, 59.69) = 5.02, *p* = 0.029). Similar to the finding for the incongruent Stroop task, performance on the congruent version of the Stroop task improved in those taking active supplementation with an ‘optimal’ diet compared to placebo but this did not reach statistical significance (*F*(1, 56.98) = 3.27, *p* = 0.076). There was also one task that found a significant treatment by time effect in favour of placebo. Those with a ‘sub-optimal’ diet had better performance on the Choice Reaction Time task after supplementation with placebo for 12 weeks compared to active supplementation (*F*(1, 57.60) = 7.02, *p* = 0.010). 

#### 3.3.2. Mood and Stress Reactivity

MMRM analyses revealed no significant treatment by time effects nor treatment by time by diet group interactions for PSS, DASS total score and subscales, or POMS total mood disturbance (see Table 3). There was also no significant benefit of active supplementation over placebo supplementation for stress reactivity in response to cognitive demand on the STAI-S, Bond-Lader, or Visual Analogue Mood Scales (data not shown). There was however a benefit in favour of active treatment over the placebo in those with an ‘optimal’ diet for lower state anxiety prior to administration of cognitive tasks (*F*(1, 64.45) = 6.12, *p* = 0.016). Similar trends were also seen in the ‘optimal’ diet group with reduced Mental Fatigue and Anxiety as measured by Visual Analogue Mood Scales (VAMS) after active supplementation, but these failed to reach statistical significance. 

#### 3.3.3. Biochemical Markers of B Vitamins

MMRM analyses revealed significant treatment by time interactions for Vitamin B1 (*F*(1, 93.65) = 17.59, *p* < 0.001), Vitamin B6 (*F*(1,124.35) = 181.00, *p* < 0.001), and Vitamin B12 (*F*(1,111.81) = 73.54, *p* < 0.001), all in favour of an increase in B vitamin status following 12 weeks of consuming active supplementation (see Table 4) as opposed to the placebo. There was a trend for Vitamin B2 to increase in the active group (*F*(1, 119.09) = 3.13, *p* = 0.079). These effects were consistent across both ‘optimal’ and ‘sub-optimal’ diet subgroups. MMRM analyses revealed no significant treatment by time interaction for homocysteine (*F*(1,119.79) = 0.002, *p* = 0.964) in the entire cohort, nor in either diet subgroup. MMRM analyses revealed also no significant treatment by time interaction for high-sensitivity C-reactive protein (*F*(1, 115.06) = 0.10, *p* = 0.754) in the entire cohort, nor in either diet subgroup. These results indicate that the B vitamin constituents of the active supplement were readily absorbed, however this did not translate to changes in homocysteine or high-sensitivity C-reactive protein.

### 3.4. Dietary Intake during Participation

Paired sample t-tests revealed that there was no significant change in total energy, macronutrient intake or micronutrient intake (including B1, B2, B6, B12) from ASA24 records between baseline and follow up. This indicated that change in biomarker status of B vitamins was unlikely due to participants changing their dietary intake across the trial. On average, participants reported higher average daily intake of alcohol at follow up (12.02 g) in comparison to baseline (7.66 g) (t = −2.69, *p* = 0.008), yet these levels were still low, equivalent to one standard drink or less per day. 

### 3.5. Treatment Blinding

While 30.5% in the active supplementation group guessed they were taking active supplementation, 18.6% of those allocated to placebo guessed they were taking active supplementation. Chi-square analysis confirmed that this difference was not significant (χ2 = 2.14, *p* = 0.192) suggesting that participants were adequately blinded to treatment allocation. 

### 3.6. Safety

Fisher Exact Tests revealed there was no difference in the occurrence of AEs between active supplementation and placebo groups (χ2 = 0.915, *p* = 0.393), suggesting that the supplement was well tolerated by participants. Analysis by System Organ Class revealed no difference in reporting across active supplementation and placebo groups for any Preferred Term. Dry mouth was reported more in the active supplementation group than in the placebo group, however this was only a trend (χ2 = 4.176, *p* = 0.058). 

## 4. Discussion

The present study aimed to evaluate the effects of a multinutrient formula containing B group vitamins, Ginkgo biloba and Bacopa monniera for memory and attention in healthy, middle-aged adults following 12 weeks supplementation. Compared to placebo, 12 weeks active supplementation with a multinutrient formulation provided no significant benefit for memory or attention when considering the whole cohort. However, analysis by diet sub-group revealed a significant benefit of supplementation for attention in participants in the ‘optimal diet’ group at baseline. Active supplementation over 12 weeks was also associated with significant increases in circulating levels of vitamins B1, B6, and B12. While B vitamins were readily absorbed (as observed by increases in B vitamin status following supplementation), supplementation was associated with relatively modest changes to mood, stress reactivity, and secondary cognitive outcomes. Further, where treatment by time effects were observed, they were largely limited to a diet subgroup, suggesting that the response to supplementation may be moderated by baseline dietary intake.

Consistent with findings of previous studies that showed an improvement in attention following multinutrient supplementation [20,21], the present study also found a significant improvement in attention. However, this was only statistically significant in those with an ‘optimal’ diet. Further inspection of the means reveals that the improvement in attention across the active supplementation is comparable with the ‘optimal’ and ‘sub-optimal’ groups. In the case of the ‘sub-optimal’ group, the improvement observed in the placebo arm (and lower baseline score) may have confounded the analysis of the sub-optimal group, and as such, the ability to detect an improvement in the overall cohort. These results should be interpreted with caution because no corrections for multiple comparisons were made on these exploratory subgroup analyses.

Unlike previous studies [15,16] we found no treatment-associated benefits on measures of memory. Importantly, these past trials were conducted in older adults than our cohort and therefore may be more sensitive to age-associated decline. Furthermore, the spatial working memory task used here is known to be particularly sensitive to age effects [47]. Given that older adults are at greater risk of memory decline, the present study may have been vulnerable to ceiling effects, with the benefits for memory not detectable until later life. 

In addition to the effects observed in those with an ‘optimal’ diet for attention, we also observed improvements associated with supplementation for the Serial Sevens subtraction task—which captures working memory and aspects of executive function—in those with a ‘sub-optimal’ diet. Inspection of the data reveals that this appeared to be driven by an increase in the number of incorrect responses in the placebo group coupled with no change in the active supplementation group. Therefore, active supplementation appeared to have a positive effect on working memory in those with a ‘sub-optimal’ diet when placed under cognitive demand. Moreover, while it appears that the placebo formulation exerted a significant improvement for choice reaction time in those with sub-optimal diets, inspection of these means revealed that this effect was largely driven by the placebo group’s poor baseline performance on this task. In fact, the relative improvement after 12 weeks resulted in a level of performance equivalent to that of the active group at baseline. This also brings to question the sensitivity of this measure. It is worth noting that, compared with the primary outcomes, choice reaction time has not been sensitive to multivitamin effects. 

The present study found that 12 weeks of active supplementation improved state anxiety in those with an ‘optimal’ diet, but it had no benefit for anxiety levels when placed under cognitive demand. These trends were again evident in the ‘optimal’ group for Mental Fatigue and Anxiety (as measured by Visual Analogue Mood Scales), again not reaching statistical significance. These modest benefits for mood are unlike previous studies which have investigated multinutrient formulas of similar concentrations and found positive mood effects in healthy adults [17,21]. Interestingly, these studies had shorter supplementation periods than the present study (≤9 weeks) but studied younger cohorts (≤50 years). As evidenced by the differential response to supplementation according to baseline diet quality, the influence of baseline participant characteristics on the outcomes of a trial cannot be understated and may explain the lack of mood effects seen in the present study. While participants were required to be free from psychiatric and mood disorders, even within the healthy range we observed differences in mood at baseline. As previously reported [55], those with an ‘optimal’ diet had significantly lower stress and mood disturbances prior to any supplementation which may have limited our ability to detect mood effects. We also observed that participants who withdrew from the study had higher levels of depressive symptoms than those who completed the study. ITT analysis controlling for baseline depression was used to overcome this, but the fact remains that the effect of supplementation in those with higher depressive symptoms was not measured. A further problem is that the magnitude of effects of multinutrient supplementation for mood are said to be small-to-moderate [12]. One might hypothesise that after controlling for baseline depression levels, the effect of supplementation in this study was negligible. 

Finally, despite significant increases in the biochemical status of Vitamin B1, B6, and B12, there was no change in homocysteine levels. This finding contradicts previous studies which have consistently reported significant reductions in homocysteine following B-complex supplementation [17,56,57]. This may be attributed to the fact the supplement under investigation did not contain folate, a cofactor which supports the clearance of homocysteine. Moreover, the baseline level of homocysteine in the active group was relatively low (9.82 µmol/L). Additionally, noteworthy is that while Vitamin B2 increased in the active group, this was a (non-significant) increase in relation to the placebo. This is likely due to the fact that we also observed an increase in status in the placebo group as the placebo formulation contained minimal amounts of Vitamin B2 (which was used to ensure adequate blinding regarding the smell and colour of urine). Further, the biochemical test for Vitamin B2 used in this study only considered the physiologically active coenzyme Flavinadeninedinucleotide (FAD). The erythrocyte glutathione reductase activation coefficient (EGRAC) is a more widely used test which considers both the physiologically inactive riboflavin and other coenzymes such as Flavinmononucleotide [58,59,60]. It is unknown whether the sensitivity of the FAD test may have limited our ability to discriminate between the greater intake of Vitamin B2 in the active group relative to the placebo.

To our knowledge, the present study was the first to recruit participants on the basis of a pre-trial dietary status. Many previous studies have hypothesised that particular subgroups within their cohort may have a heightened response to supplementation (e.g., those with sub-optimal nutrient status, those with poor mood). However, due to limited methodology, there are often no means of identifying these individuals. Furthermore, post hoc subgroup analyses are often limited by a restricted range in both nutrient status and dietary intakes across participants. By recruiting 50% of participants with a ‘sub-optimal’ diet and 50% with an ‘optimal’ diet, and then randomizing accordingly, the present study was able to conduct a series of a priori analyses which explored the differential effects of supplementation according to their existing diet quality. In the absence of a main effect of treatment by time, a number of significant three-way interactions (treatment by time by diet group) were detected, indicating that supplementation may interact with baseline diet quality. As the study was powered to detect effects in the primary outcomes, these subsequent diet subgroup analyses were exploratory and, therefore, require replication. Nevertheless, these findings highlight the importance of understanding and collecting baseline characteristics (including diet quality) from participants, as such variables may modulate the effects of nutraceutical supplementation. 

Active supplementation appeared to provide a benefit for attentional tasks. Although contrary to expectations, this was only statistically significant in individuals with an ‘optimal’ diet but not those with a ‘sub-optimal’ diet. While we anticipated that those with a ‘sub-optimal’ diet would have poorer baseline nutrient status, and therefore an enhanced response from consuming more nutrients, we in fact saw the opposite effect with those with a pre-existing ‘optimal’ diet achieving attentional benefits from supplementation in comparison to placebo. While unexpected, the beneficial effects within the ‘optimal’ diet group supports the concept of ‘co-nutrient optimisation’ which purports that the interdependence of nutrients are an essential consideration when studying nutrient effects [39]. In other words, the benefit derived from supplementing with nutrient X is dependent on the optimisation of nutrients W, Y and Z [39,61]. Deficiencies in other nutrient biomarkers may prevent the uptake and absorption of the nutrient under investigation and thus limit the benefits which are measurable within a clinical trial setting. This interdependency of nutrients is well accepted in nutrition [62,63,64]. In particular, the metabolism of B vitamins is interrelated with previous studies reporting that riboflavin supplementation not only corrects riboflavin insufficiency, but assists the uptake of vitamin B6 [60]. In the case of the present study individuals with an ‘optimal’ diet may see enhanced effects of supplementation because they were also consuming a diet which is rich in nutrients that assist the metabolism of B group vitamins and herbal constituents in the supplement. Comparatively, those with a ‘sub-optimal’ diet may be insufficient in several nutrients (including ones not included in the supplement) and therefore the effects of B vitamins and herbal supplementation may be limited. Previous studies of multinutrient supplements have observed this interdependence of nutrients. An RCT in healthy adults providing omega-3 and multivitamins both individually and in combination found that long-chain omega-3 index only improved in those receiving fish oil and multivitamins in combination, suggesting that the multinutrient aided the incorporation of polyunsaturated fatty acids into red blood cells [65]. Similarly, supplementation with a B vitamin formula slowed cognitive decline and brain atrophy only if participants had high omega-3 status at baseline [66,67]. Most recently, a longitudinal study in Australia (*n* = 69,990) explored the combined effect of diet quality and dietary supplements on obesity and cardiovascular disease [68]. Similar to the findings of the present study, they reported that a long-term healthy diet combined with calcium supplementation was associated with lower incidence of obesity—but these associations were not evident in those with an unhealthy diet who took similar supplements. 

Within this context, it is also important to note that in comparison to individual B vitamin interventions, the concentrations of B vitamins within this investigational product were relatively low. This may also provide explanation as to why we saw benefits in those who had a relatively high intake of healthy foods but not in those in the ‘sub-optimal’ diet group as the dosages of B vitamins may not be large enough to rectify the detriments associated with consuming a nutrient-poor diet. The observed benefits in those with an ‘optimal’ diet may be the interaction with aspects of diet (macronutrient intake, nutrients other than B vitamins and non-nutrients) which when combined, provide a benefit to the brain. Magnesium is known to be an important cofactor for Vitamin B6 absorption [69] and we have already reported that the ‘optimal’ group had a higher intake of magnesium than the ‘sub-optimal’ group. The investigational product in the present study did not provide the full range of B vitamins—notably the absence of folate. Moreover, it only contained two herbal constituents (Bacopa monnieri and Ginkgo biloba)—both of which were in low doses when contrasted to studies which have previously shown cognitive enhancing effects [36]. Comparatively, in the previously mentioned study from our group which observed improvements in the incongruent Stroop task following supplementation, the corresponding investigational product comprised comparable doses of B vitamins, but an additional 48 vitamins, minerals, and herbal constituents. One may hypothesise that these other nutrients may have enhanced the effect of B vitamins, in the same manner that was observed in the ‘optimal’ diet group for the present study. Together, these findings suggest that supplementation with B group vitamins, Bacopa monnieri and Ginkgo biloba has the potential to improve attention, but these benefits may be conditional on the support of a healthy diet. Furthermore, this finding provides support for holistic diet approaches which argue that the synergy of whole diet, or providing a broad-spectrum multivitamin, may be more effective at rectifying nutritional deficiencies than when supplementing individual nutrients [37,70,71,72,73].

The present study has a number of limitations. First, all nutritional interventions can never truly have a ‘no exposure’ group as participants will inevitably consume nutrients within their habitual diet and these levels may differ between individuals [74]. We attempted to overcome this by measuring biochemical markers of B vitamins pre- and post-intervention as well as analysing micronutrient intakes across the study to ensure participants maintained a habitual diet. More extensive assays of nutrient biomarker status will be useful for future research. Second, participants who completed the study had fewer depressive symptoms than those who withdrew from the study. While this was incorporated into the analyses, it raises the possibility that individuals who may have benefited most from supplementation did not complete follow up assessments. Future studies targeting individuals with elevated distress or poorer mood at baseline may yield different results or see more pronounced nutrient effects [13]. Third, a very small proportion of participants had B vitamin insufficiency at baseline. In a trial of this nature which considered a multinutrient, using overall diet quality was deemed appropriate to capture those who may be insufficient across a broad range of nutrients, with the aim of also capturing those with poor B vitamin status. However, as the Australian version of the DST was not sensitive to all B-group vitamins in this cohort [55], the findings of this trial must be interpreted within the context of B vitamin sufficiency. Future studies interested in individual nutrients may benefit from the use of more refined screening measures which are specific to individual nutrients and use biochemical screens for nutrients of interest prior to enrolment. Fourth, the active supplement comprised multiple B vitamins and herbal constituents (i.e., Bacopa monniera and Ginkgo biloba). It is therefore difficult to discern the direct impact of each of these constituents on the behavioural outcomes. The herbal constituents may act through a mechanism which is independent of B vitamins and baseline diet intake. This study chose to focus on shifts in B vitamin levels due to the herbal doses being relatively low compared to other studies [36] and because there are no biomarker indicators of Bacopa monniera, and Ginkgo biloba absorption, however, we are unable to quantify how this may have influenced the results. Moreover, the placebo formulation contained minimal amounts of Vitamin B2 (for blinding purposes), which resulted in the placebo group observing increases in this biomarker. While we would not anticipate that this minute amount of Vitamin B2 could account for the increased performance in attention in the placebo, sub-optimal diet group, this unexpected, ‘placebo’ effect may have confounded the results in the primary outcome and our ability to detect a benefit in the overall cohort. Thus, the possibility that the placebo formulation confounded the results cannot be ruled out. Fifth, the majority of the study sample was well educated and Caucasian, reducing the generalisability of the findings. Finally, while this study was powered for the co-primary outcomes (follow up data for *n* = 116), all analysis beyond this (including dietary subgroup analysis) was considered exploratory. The findings of this study therefore require replication to ensure that they are not the result of Type I error. 

This study is strengthened by adequate blinding and a novel approach to recruitment to include a broad variety of participants in terms of their baseline diet quality. Randomization by dietary status allowed for a priori analysis by diet subgroup and the revelation of differential supplement effects according to baseline diet. The cognitive batteries used were domain specific and allowed us to capture the range of cognitive functions that are particularly sensitive to nutraceutical interventions. A rigorous statistical approach was adopted, which considered baseline differences, attrition analysis and subgroup analysis–all factors which are often overlooked in nutritional interventions [41]. By monitoring dietary intake across the study we were able to ascertain that the change in biomarker status of B vitamins was attributable to the investigational supplement rather than participants changing their diet over the course of the intervention period. Finally, the study of middle-age adults is under researched but offers opportunity to study the effect of lifestyle interventions in a life stage where the effects may be more pronounced [38]. In a rapidly ageing population, early interventions such as the one used in this trial for a middle-aged group (40–65 years) may prove critical to slowing the ageing effects on the brain. 

This trial contributes to the growing evidence for combination formulas with high concentrations of B vitamins to improve nutrient status and cognition in healthy, non-clinical populations. Analysis of nutrient biomarkers support the uptake of B vitamins and analysis of adverse events demonstrated it was safe for use. A critical finding of this trial was that baseline diet interacted with supplementation, revealing the differential effects of supplementation according to existing diet quality. This requires replication and baseline dietary status should be controlled for in future studies. Future research should also explore the interaction between gender and multivitamin supplementation. If the results are replicated in future research, it will have important ramifications for the field of nutrition science and may provide explanation for the equivocal nature of supplementation research thus far. Previous null findings may in fact be a by-product of not characterising participants adequately prior to intervention and thus we see the variation in baseline diet quality overshadowing any intervention effects. It also emphasises the issue of inter-individual variation in response to supplementation, whereby pre-existing diet determines an individual’s response to supplementation. Understanding the baseline characteristics of participants (or patients) will also help guide the development of personalised intervention strategies within the growing field of ‘precision nutrition’ [75].

Finally, these findings also have implications for the supplementation industry regarding their target consumers and formulation of products. Our findings suggest that supplements should not be used in replacement of a healthy diet and rather used in conjunction with healthy foods. Future studies should consider the interaction of B vitamins with baseline diet and other micronutrients including omega-3. In line with this, there is increasing consensus that B-group vitamins work in synergy [37] and thus the absence of folate in this supplement may have acted as a ‘limiting’ nutrient within their complex interlinked metabolic cycles. In order to determine the true efficacy of multinutrient supplements, future studies should consider providing broad spectrum B-complexes, including folate, to ensure adequacy across all B group vitamins.

## 5. Conclusions

The present study provides support for the use of multinutrient formula containing B group vitamins and herbals to improve nutrient biomarkers in healthy, middle-aged adults. Contrary to expectations, improvement in attentional tasks were statistically significant in participants with an ‘optimal’ diet prior to supplementation, which supports the concept of ‘co-nutrient optimisation’ and interdependency of nutrients. The supplement provided no significant benefit to memory as measured by a spatial working memory task. The differential effects based on baseline dietary status evident in this study may provide explanation for the overall equivocal nature of previous research within nutrition science. Extensive characterisation of participants prior to supplementation is therefore warranted and this may assist in the identification of those who are most likely to benefit from nutraceutical interventions. 

## Figures and Tables

**Figure 1 nutrients-14-05079-f001:**
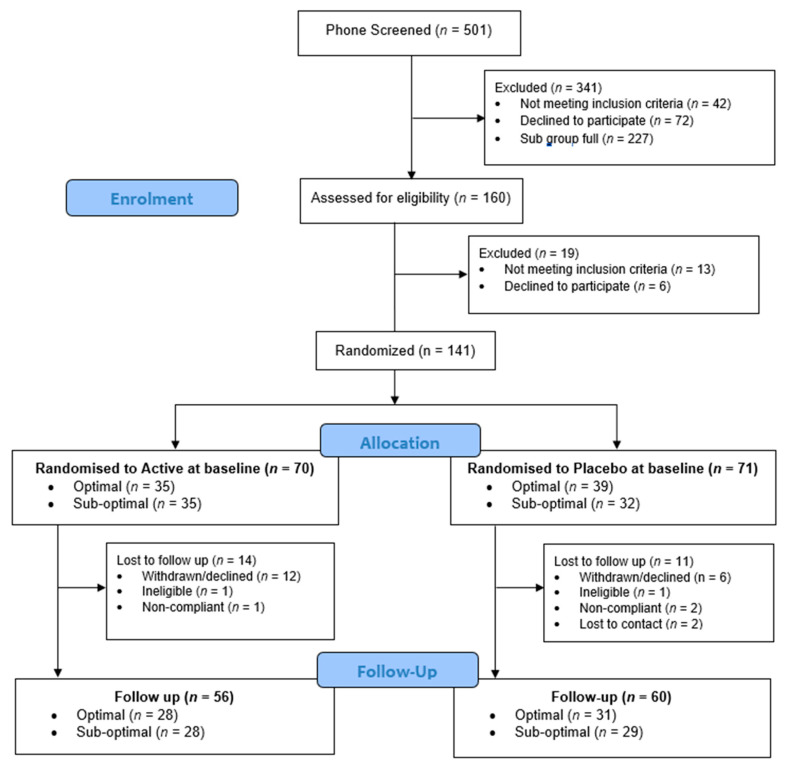
CONSORT diagram.

**Figure 2 nutrients-14-05079-f002:**
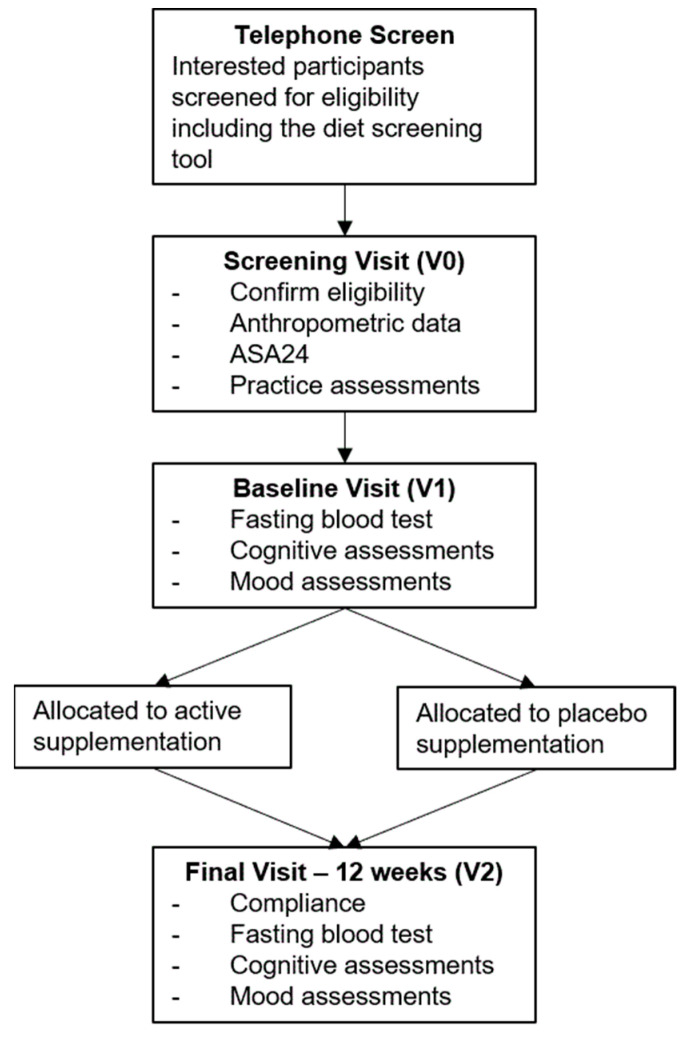
Outline of Study Assessments.

**Figure 3 nutrients-14-05079-f003:**
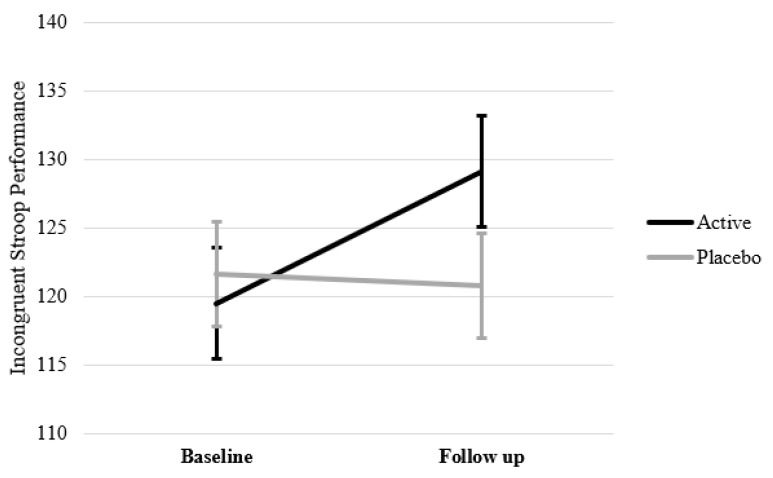
Incongruent Stroop Performance (Attention) in individuals with an ‘optimal’ diet at baseline and follow up (12 weeks) after adjusting for age, gender, education, BMI, BDI-II and brachial blood pressure.

**Table 1 nutrients-14-05079-t001:** Sample characteristics for entire cohort and by treatment allocation at baseline.

		Whole Sample		Active		Placebo		
	*n*	*M*	*SD*	*M*	*SD*	*M*	*SD*	*p*
Age	141	52.84	6.87	53.02	6.78	52.67	7.01	0.763
MMSE	141	29.31	0.95	29.30	0.97	29.34	0.94	0.813
Education years	141	16.94	3.36	17.23	3.75	16.66	2.93	0.319
BMI	141	27.26	5.22	28.19	5.59	26.35	4.67	0.036 *
Waist Circumference	140	95.91	14.77	97.97	15.74	93.85	13.54	0.099
Hip Circumference	140	107.21	9.15	105.35	10.03	106.06	8.08	0.139
BDI	141	4.00	4.37	3.97	4.40	4.03	4.36	0.939
Diet quality	141	62.10	13.00	60.40	12	63.77	13.31	0.124
NART	141	37.13	5.38	36.79	5.60	37.45	5.17	0.465
STAI-trait	141	31.48	7.42	31.43	7.65	31.54	7.25	0.932
Brachial systolic	141	119.28	12.36	120.30	12.67	118.28	12.05	0.334
Brachial diastolic	141	73.92	8.47	73.84	9.08	74.00	7.90	0.913
Vitamin B1	94	126.60	33.80	126.22	36.85	127.00	30.56	0.912
Vitamin B2	135	233.36	42.70	233.91	38.68	232.79	46.82	0.879
RBC Folate	62	1242.50		1285.15	152.06	1211.69	314.75	0.464
Vitamin B6	139	116.85	118.54	114.06	108.10	119.60	128.73	0.833
Vitamin B12	139	321.09	114.11	329.33	99.74	312.96	126.89	0.271
Homocysteine	139	10.28	2.84	9.82	2.55	10.73	3.05	0.062
High-sensitivity C-reactive protein	135	1.77	2.98	2.12	3.56	1.43	2.28	0.180

Variables were compared across active treatment and placebo groups using t-tests. * *p* < 0.05.

**Table 2 nutrients-14-05079-t002:** Means, standard deviations, *F* value for treatment by time interaction and significance values from MMRMA analysis for cognitive outcomes at baseline and 12 weeks for active and placebo groups.

		Active				Placebo					
Task	Subgroup	*n*	Baseline *M (SD)*	12 Weeks *M (SD)*	Δ	*n*	Baseline *M (SD)*	12 Weeks *M (SD)*	Δ	*F*	*p*
Spatial Working Memory Performance	All	70	89.61 (25.64)	92.28 (28.54)	2.67	71	90.96 (28.60)	95.50 (32.24)	4.54	0.53	0.470
	Optimal	35	87.63 (27.38)	88.04 (29.51)	0.41	39	87.18 (29.23)	87.31 (29.53)	0.13	0.08	0.772
	Sub-Optimal	35	91.59 (24.02)	96.52 (27.41)	4.93	32	95.57 (27.57)	104.25 (33.20)	8.68	0.54	0.467
Stroop Incongruent Performance	All	69	121.40 (22.64)	130.67 (21.09)	9.27	70	119.01 (23.57)	124.38 (25.90)	5.37	1.89	0.171
	Optimal	34	120.59 (24.32)	130.38 (18.05)	9.79	38	120.88 (23.22)	120.66 (26.53)	−0.22	4.94	0.030 *
	Sub-Optimal	35	122.19 (21.20)	130.96 (24.08)	8.77	32	116.79 (24.15)	128.35 (25.06)	11.56	0.37	0.544
Simple Reaction Time	All	70	380.18 (35.34)	380.23 (51.02)	0.05	71	375.82 (57.09)	375.37 (51.07)	−0.45	0.15	0.697
	Optimal	35	384.60 (56.34)	379.90 (56.47)	−4.70	39	379.71 (59.60)	380.96 (54.68)	1.25	0.99	0.323
	Sub-Optimal	35	375.75 (52.71)	380.56 (45.97)	4.81	32	371.08 (54.44)	369.40 (47.12)	−1.68	0.26	0.609
Choice Reaction Time Performance	All	70	220.35 (32.79)	217.47 (32.60)	−2.88	71	211.97 (29.61)	216.67 (31.69)	4.70	2.02	0.158
	Optimal	35	217.69 (36.03)	215.71 (34.94)	−1.98	39	217.24 (28.28)	214.80 (30.81)	−2.44	0.72	0.399
	Sub-Optimal	35	223.01 (29.47)	219.23 (30.63)	−3.78	32	205.55 (30.36)	218.67 (33.04)	13.12	7.02	0.010 *
Immediate Recognition Performance	All	70	83.34 (16.46)	87.06 (16.70)	3.72	70	86.40 (19.95)	90.02 (19.79)	3.62	0.63	0.430
	Optimal	35	84.53 (18.06)	86.38 (15.61)	1.85	39	84.24 (19.48)	86.39 (20.51)	2.15	0.00	0.998
	Sub-Optimal	35	82.16 (14.86)	87.73 (17.99)	5.57	31	89.12 (20.52)	93.90 (18.54)	4.78	1.21	0.276
Delayed Recognition Performance	All	70	72.30 (16.06)	68.07 (15.23)	−4.23	71	73.48 (17.16)	71.77 (16.86)	−1.71	0.85	0.359
	Optimal	35	76.67 (17.72)	68.95 (13.57)	−7.72	38	72.59 (17.71)	68.20 (16.12)	−4.39	0.84	0.364
	Sub-Optimal	35	67.93 (13.06)	67.19 (16.93)	−0.74	32	74.53 (16.70)	75.59 (17.06)	1.06	0.18	0.670
Stroop Congruent Performance	All	70	142.53 (20.82)	148.42 (21.80)	5.89	71	142.74 (22.28)	144.65 (18.92)	1.91	2.55	0.11
	Optimal	35	138.29 (22.07)	144.97 (21.17)	6.68	39	142.87 (22.02)	143.90 (18.79)	1.03	3.27	0.076 ^
	Sub-Optimal	35	146.78 (18.85)	151.88 (22.25)	5.10	32	142.58 (22.94)	145.46 (19.36)	2.88	0.23	0.631
Contextual Memory Performance	All	70	83.05 (22.53)	90.02 (23.30)	6.97	70	86.01 (23.00)	93.32 (21.33)	7.31	0.04	0.853
	Optimal	35	83.71 (27.83)	88.98 (26.97)	5.27	38	83.41 (25.33)	89.22 (23.51)	5.81	1.79	0.186
	Sub-Optimal	35	82.40 (15.95)	91.07 (19.39)	8.67	32	89.10 (19.84)	97.70 (18.11)	8.61	1.16	0.285
Serial Threes correct (#)	All	70	31.75 (13.83)	33.10 (13.63)	1.35	71	31.36 (13.85)	33.71 (14.36)	2.35	1.16	0.284
	Optimal	35	30.75 (12.85)	30.37 (12.65)	−0.38	39	30.61 (11.91)	32.63 (11.85)	2.02	1.49	0.227
	Sub-Optimal	35	32.75 (14.86)	35.73 (14.22)	2.98	32	32.27 (16.05)	34.85 (16.79)	2.58	0.04	0.835
Serial Threes incorrect (#)	All	70	2.60 (2.16)	2.84 (2.53)	0.24	71	2.66 (2.10)	3.18 (2.42)	0.52	0.23	0.633
	Optimal	35	2.63 (1.77)	3.14 (3.00)	0.51	39	2.80 (2.36)	3.39 (2.41)	0.59	0.01	0.957
	Sub-Optimal	35	2.58 (2.52)	2.55 (1.98)	−0.03	32	2.48 (1.75)	2.95 (2.45)	0.47	0.43	0.515
Serial Sevens correct (#)	All	70	20.86 (10.62)	21.91 (11.34)	1.05	70	20.80 (11.43)	22.02 (12.46)	1.22	0.02	0.893
	Optimal	35	20.16 (9.92)	20.10 (10.79)	−0.06	39	19.79 (10.04)	21.63 (11.16)	1.84	2.05	0.157
	Sub-Optimal	35	21.55 (11.38)	23.65 (11.78)	2.10	31	22.08 (13.03)	22.43 (13.90)	0.35	2.61	0.112
Serial Sevens incorrect (#)	All	70	3.10 (1.76)	3.24 (2.23)	0.14	70	2.96 (2.11)	3.87 (3.30)	0.91	3.15	0.078 ^
	Optimal	35	3.04 (1.84)	3.59 (2.19)	0.55	39	3.28 (2.23)	4.17 (3.83)	0.89	0.26	0.610
	Sub-Optimal	35	3.15 (1.70)	2.90 (2.25)	−0.25	31	2.56 (1.91)	3.55 (2.66)	0.99	5.02	0.029 *
RVIP Performance	All	68	10.23 (4.96)	11.20 (5.14)	0.97	68	10.56 (4.47)	11.79 (4.74)	1.23	0.05	0.824
	Optimal	33	10.76 (5.63)	11.78 (5.77)	1.02	37	10.62 (4.38)	11.97 (4.55)	1.35	0.04	0.840
	Sub-Optimal	35	9.74 (4.27)	10.70 (4.57)	0.96	31	10.50 (4.65)	11.62 (4.99)	1.12	0.35	0.554
RVIP False Alarms (#)	All	68	8.57 (5.77)	9.47 (10.14)	0.90	68	8.52 (6.42)	7.63 (5.70)	−0.89	0.02	0.895
	Optimal	33	8.99 (7.55)	8.19 (10.98)	−0.80	37	8.57 (6.34)	7.68 (5.98)	−0.89	3.61	0.063 ^
	Sub-Optimal	35	8.17 (3.42)	10.56 (9.43)	2.39	31	8.46 (6.62)	7.57 (5.53)	−0.89	2.28	0.137
RVIP Missed (#)	All	68	21.87 (8.12)	20.37 (8.69)	−1.50	68	20.91 (7.87)	18.87 (8.24)	−2.04	0.28	0.600
	Optimal	33	21.44 (9.18)	19.40 (9.64)	−2.04	37	20.90 (7.55)	18.76 (7.57)	−2.14	0.21	0.651
	Sub-Optimal	35	22.46 (7.08)	21.20 (7.88)	−1.26	31	20.92 (8.35)	18.98 (8.97)	−1.94	0.07	0.793

MMRMA models were controlled for age, gender, education, BMI, BDI-II and brachial blood pressure. Δ indicates mean change between baseline and 12 weeks. F indicates F value for treatment x time interaction. Performance on each task was computed as accuracy (%) divided by speed of response (in milliseconds) where higher scores were equivalent to better performance. * *p* < 0.05, ^ *p* < 0.10.

**Table 3 nutrients-14-05079-t003:** Means, standard deviations, *F* value for treatment by time interaction and significance values from MMRMA analysis for mood outcomes at baseline and 12 weeks for active and placebo groups.

		Active				Placebo					
Task	Subgroup	*n*	Baseline *M (SD)*	12 Weeks *M (SD)*	Δ	*n*	Baseline *M (SD)*	12 Weeks *M (SD)*	Δ	*F*	*p*
Perceived Stress Scale	All	70	11.39 (5.82)	11.98 (6.45)	0.59	71	11.35 (5.93)	11.23 (6.15)	−0.12	0.26	0.611
	Optimal	35	10.77 (6.15)	10.89 (5.70)	0.12	39	10.85 (6.12)	11.10 (6.25)	0.25	0.33	0.569
	Sub-Optimal	35	12.00 (5.50)	13.07 (7.05)	1.07	32	11.97 (5.73)	11.38 (6.15)	−0.59	1.22	0.273
DASS Total	All	70	6.27 (6.79)	6.93 (6.80)	0.66	71	5.96 (5.47)	5.28 (6.80)	−0.68	0.81	0.371
	Optimal	35	5.97 (6.69)	5.93 (6.93)	−0.04	39	5.90 (5.88)	4.48 (4.25)	−1.42	0.01	0.927
	Sub-Optimal	35	6.57 (6.98)	7.93 (6.65)	1.36	32	6.03 (5.02)	6.14 (8.75)	0.11	1.36	0.247
DASS Stress	All	70	3.26 (3.19)	3.55 (3.42)	0.29	71	3.00 (2.69)	2.62 (3.06)	−0.38	0.96	0.329
	Optimal	35	3.00 (3.08)	2.68 (3.14)	−0.32	39	3.03 (2.84)	2.29 (2.19)	−0.74	0.01	0.961
	Sub-Optimal	35	3.51 (3.33)	4.43 (3.51)	0.92	32	2.97 (2.53)	2.97 (3.79)	0.00	2.04	0.159
DASS Anxiety	All	70	1.06 (1.65)	1.45 (1.72)	0.39	71	0.85 (1.67)	0.88 (1.67)	0.03	0.17	0.685
	Optimal	35	1.14 (1.65)	1.50 (1.93)	0.36	39	1.05 (1.99)	0.84 (1.13)	−0.21	0.06	0.816
	Sub-Optimal	35	0.97 (1.67)	1.39 (1.50)	0.42	32	0.59 (1.16)	0.93 (2.12)	0.34	0.07	0.796
DASS Depression	All	70	1.96 (2.88)	1.93 (2.88)	−0.03	71	2.11 (2.46)	1.78 (2.92)	−0.33	0.31	0.582
	Optimal	35	1.83 (2.77)	1.75 (3.09)	−0.08	39	1.82 (2.14)	1.35 (2.04)	−0.47	0.01	0.973
	Sub-Optimal	35	2.09 (3.02)	1.39 (1.50)	−0.70	32	2.47 (2.81)	2.24 (3.62)	−0.23	0.84	0.362
POMS Total Mood Disturbance	All	70	7.46 (25.09)	5.38 (24.26)	−2.08	71	3.58 (21.30)	−0.87 (18.07)	−4.45	0.00	0.999
	Optimal	35	3.57 (23.63)	−1.18 (22.28)	−4.75	39	1.69 (23.96)	−3.03 (17.60)	−4.72	1.18	0.282
	Sub-Optimal	35	11.34 (26.23)	11.93 (24.77)	0.59	32	5.88 (17.64)	1.45 (18.58)	−4.43	1.08	0.303
STAI-S (State Anxiety)	All	70	27.87 (7.11)	27.84 (8.09)	−0.03	71	28.46 (6.40)	29.62 (8.55)	1.16	0.91	0.341
	Optimal	35	28.74 (8.20)	26.14 (6.91)	−2.60	39	28.69 (7.35)	29.68 (8.41)	0.99	6.12	0.016 *
	Sub-Optimal	35	27.00 (5.80)	29.54 (8.93)	2.54	32	28.19 (5.13)	29.55 (8.84)	1.36	0.68	0.412
Bond Lader Alertness	All	70	70.87 (16.09)	71.17 (15.05)	0.30	71	71.23 (14.72)	73.06 (14.65)	1.83	0.00	0.989
	Optimal	35	70.66 (16.92)	73.08 (14.26)	2.42	39	69.09 (16.36)	71.43 (14.28)	2.34	0.19	0.668
	Sub-Optimal	35	71.07 (15.46)	69.26 (15.83)	−1.81	32	73.86 (12.18)	74.80 (15.08)	0.94	0.06	0.806
Bond Lader Calmness	All	70	75.42 (14.09)	71.86 (16.60)	−3.56	71	72.18 (13.57)	73.10 (15.16)	0.92	2.04	0.155
	Optimal	35	75.24 (15.27)	75.98 (15.45)	0.74	39	69.62 (13.45)	72.50 (13.74)	2.88	0.37	0.548
	Sub-Optimal	35	75.60 (13.02)	67.73 (16.94)	−7.87	32	75.30 (13.27)	73.74 (16.76)	−1.56	1.76	0.325
Bond Lader Contentedness	All	70	78.03 (15.24)	76.31 (13.81)	−1.72	71	77.41 (12.83)	74.95 (14.79)	−2.46	0.57	0.452
	Optimal	35	77.54 (16.25)	77.33 (14.09)	−0.21	39	75.64 (13.60)	74.29 (14.30)	−1.35	0.46	0.498
	Sub-Optimal	35	78.53 (14.38)	75.29 (13.71)	−3.24	32	79.58 (11.68)	75.66 (15.52)	−3.92	0.19	0.665
Bond Lader Stress	All	70	20.94 (21.39)	21.41 (18.77)	0.47	71	19.25 (16.52)	25.25 (19.87)	6.00	1.94	0.166
	Optimal	35	21.17 (21.61)	18.93 (17.66)	−2.24	39	22.95 (18.55)	27.19 (17.82)	4.24	1.57	0.215
	Sub-Optimal	35	20.71 (21.47)	23.89 (19.84)	3.18	32	14.75 (12.50)	23.17 (21.98)	8.42	0.51	0.477
Bond Lader Anxiety	All	70	16.94 (15.37)	21.36 (19.37)	4.42	71	20.35 (18.48)	24.07 (20.10)	3.72	0.04	0.847
	Optimal	35	18.37 (17.03)	16.86 (17.78)	−1.51	39	23.95 (19.93)	26.55 (18.35)	2.60	2.85	0.097 ^
	Sub-Optimal	35	15.51 (13.60)	25.86 (20.15)	10.35	32	15.97 (15.75)	21.41 (21.82)	5.44	1.25	0.269
Bond Lader Mental Fatigue	All	70	31.71 (24.10)	27.84 (19.72)	−3.87	71	28.83 (21.79)	28.70 (19.57)	−0.13	1.04	0.311
	Optimal	35	33.91 (25.70)	24.29 (17.39)	−9.62	39	31.46 (23.61)	31.74 (18.96)	0.28	2.89	0.094 ^
	Sub-Optimal	35	29.51 (22.55)	31.39 (21.52)	1.88	32	25.63 (19.23)	25.45 (20.02)	−0.18	0.04	0.848
Bond Lader Concentration	All	70	67.80 (19.62)	65.95 (18.13)	−1.85	71	69.96 (18.48)	69.73 (15.94)	−0.23	0.04	0.847
	Optimal	35	65.57 (20.33)	66.25 (17.94)	0.68	39	67.41 (20.39)	67.71 (15.55)	0.30	0.32	0.574
	Sub-Optimal	35	70.03 (18.92)	65.64 (18.64)	−4.39	32	73.06 (15.61)	71.90 (16.35)	−1.16	0.03	0.862
Bond Lader Mental Stamina	All	70	68.30 (19.04)	65.23 (18.09)	−3.07	71	68.59 (18.16)	69.28 (15.69)	0.69	0.45	0.506
	Optimal	35	66.66 (20.43)	67.04 (18.28)	0.38	39	65.69 (19.63)	67.77 (14.85)	2.08	0.01	0.935
	Sub-Optimal	35	69.94 (17.68)	63.43 (18.05)	−6.51	32	72.13 (15.78)	70.90 (16.65)	−1.23	0.53	0.469

MMRMA models were controlled for age, gender, education, BMI, BDI-II and brachial blood pressure. Δ indicates mean change between baseline and 12 weeks. F indicates F value for treatment x time interaction. The values presented for Stress Reactivity (STAI-S, Bond Lader scales) are mean scores prior to any cognitive tasks. * *p* < 0.05, ^ *p* < 0.10.

**Table 4 nutrients-14-05079-t004:** Means, standard deviations, *F* value for treatment by time interaction and significance values from MMRMA analysis for blood levels of vitamin B and homocysteine at baseline and 12 weeks for active and placebo groups.

		Active				Placebo					
Measure	Subgroup	*n*	Baseline *M (SD)*	12 Weeks *M (SD)*	Δ	*n*	Baseline *M (SD)*	12 Weeks *M (SD)*	Δ	*F*	*p*
Vitamin B1 (nmol/L)	All	49	126.22 (36.85)	175.72 (52.03)	49.5	45	127.00 (30.56)	130.08 (50.86)	3.08	17.59	**0.000 ***
	Optimal	25	126.68 (32.16)	183.63 (56.08)	56.95	27	129.70 (33.26)	140.28 (64.21)	10.58	4.90	**0.031 ***
	Sub-Optimal	24	125.75 (41.87)	169.46 (48.88)	43.71	18	122.94 (26.39)	120.27 (31.79)	−2.67	20.59	**0.002 ***
Vitamin B2 (nmol/L)	All	69	233.91 (38.68)	282.30 (46.00)	48.39	66	232.79 (46.82)	261.05 (47.56)	28.26	3.13	0.079 ^
	Optimal	35	231.29 (36.73)	286.96 (42.94)	55.67	35	230.00 (33.38)	263.79 (55.76)	33.79	3.34	0.072 ^
	Sub-Optimal	34	236.62 (40.97)	277.96 (49.05)	41.34	31	235.94 (58.88)	258.11 (37.67)	22.17	0.48	0.490
Vitamin B6 (nmol/L)	All	69	114.06 (108.10)	608.71 (250.14)	494.65	70	119.60 (128.73)	121.24 (139.91)	1.64	181.00	**0.000 ***
	Optimal	35	137.29 (145.09)	612.21 (278.16)	474.92	38	139.79 (164.18)	153.96 (190.24)	14.17	42.60	**0.000 ***
	Sub-Optimal	34	90.15 (35.09)	605.96 (231.70)	515.81	32	95.63 (60.15)	89.77 (47.99)	−5.86	263.95	**0.000 ***
Vitamin B12 (pmol/L)	All	69	329.33 (99.74)	510.57 (195.90)	181.24	70	312.96 (126.89)	307.73 (140.82)	−5.23	73.54	**0.000 ***
	Optimal	35	316.94 (90.03)	510.50 (185.34)	193.56	39	299.79 (124.34)	299.33 (86.87)	−0.46	39.86	**0.000 ***
	Sub-Optimal	34	342.09 (108.71)	510.64 (208.62)	168.55	31	329.52 (130.15)	316.13 (180.70)	−13.39	34.09	**0.000 ***
Homocysteine (μmol/L)	All	69	9.82 (2.55)	9.68 (2.93)	−0.14	70	10.73 (3.05)	10.55 (2.97)	−0.18	0.00	0.964
	Optimal	35	9.41 (2.37)	9.70 (2.77)	0.29	39	10.58 (2.81)	10.73 (2.72)	0.15	0.00	0.938
	Sub-Optimal	34	10.25 (2.69)	9.67 (3.14)	−0.58	31	10.92 (3.36)	10.37 (3.26)	−0.55	0.00	0.982
	All	66	1.02 (0.09)	0.98 (0.10)	−0.04	69	1.01 (0.9)	1.00 (0.10)	−0.01	0.10	0.754
High-sensitivity C-reactive protein (mg/L)	Optimal	34	0.95 (0.13)	0.97 (0.14)	0.02	39	0.98 (0.12)	0.93 (0.13)	−0.05	0.25	0.620
	Sub-Optimal	32	1.06(0.13)	0.95 (0.14)	−0.11	30	1.09(0.14)	1.12(0.14)	0.03	0.54	0.467

MMRMA models were controlled for age, gender, education, BMI, BDI-II and brachial blood pressure. Δ indicates mean change between baseline and 12 weeks. F indicates F value for treatment by time interaction. * *p* < 0.05, ^ *p* < 0.10.

## Data Availability

The data analyzed in this study is subject to the following licenses/restrictions: Need ethical clearance and industry sponsor approval to access data. Requests to access these data sets should be directed to the corresponding author.

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
