# Peer review of "Investigating the Effects of a Multinutrient Supplement on Cognition, Mood and Biochemical Markers in Middle-Aged Adults with ‘Optimal’ and ‘Sub-Optimal’ Diets: A Randomized Double Blind Placebo Controlled Trial"

_nutrients, 2022, doi:10.3390/nu14235079_

Round 1

Reviewer 1 Report

This is a comprehensive and well-designed study. The interpretation of findings and discussion of results are measured and take into consideration the limitations of the trial. Minor points for clarification are detailed below.

Could you add the age criteria to the Participant section

Not sure I understand speed-accuracy calculation, would it not be preferable to know the effects on speed and accuracy separately? They might be opposite effects, but this detailed information is useful rather than the composite, which wouldn’t show up this information.

Table 2 seems to present speed or accuracy data depending on the task. It seems that the result for incongruent Stroop is based on speed only, is that correct?

Clarify outcomes for RVIP – what is performance? Is speed not measured?

Was only TMD included for POMS?

Will inflammation markers be published elsewhere? They are referred to in the clinical trial reg but not in the paper.

Please include data for all outcomes – speed and accuracy – in Table 2

Figure 3 - please indicate that these are adjusted means in the figure title. The Y axis should be labelled with the outcome variable – ms?

Does the co-nutrient optimisation argument stack up when you have no indication of differences in absorption for the biomarkers measured? It would be worth referring to biomarker levels when discussing impact of baseline diet on absorption.

Is there any reason to think that optimal diet would impact on response to herbal extracts? What is the proposed mechanism for this? If not, how do the herbal extracts fit in? Might be worth adding a comment on gut microbiota.

Any idea how many participants came from online recruitment and how many came from posters around uni? May be worth commenting on the selective participant base.

Line 331 – the supplementation in reference 18 did not include an herbal extract

Line 635 – anxiety and mental fatigue assessed by Bond-Lader are referred to. Please check this as these outcomes aren’t produced by these scales

Some grammatical errors in the Discussion, worth rechecking this

Reviewer 2 Report

Review of “Investigating the Effects of a Multinutrient Supplement on Cognition, Mood and Biochemical Markers in Middle-Aged Adults with ‘Optimal’ and ‘Sub-optimal’ Diets: A Randomized Double Blind Placebo Controlled Trial”

1.     Line 84 stated that there is inadequate evidence for Bacopa monnieri's effects in cognitive domains depending on (PMID: 22747190). Since a recent systematic review showed that Bacopa monnieri enhances language, learning, and memory, I believe you should follow up on new research and include it in the article (PMID: 34978226).

2.     According to what you stated in the introduction, the improvement would be after 12 weeks? It was stated that the duration could be the reason for the lack of significant results in the referenced studies (lines75, 76); other studies took 16 weeks to complete their research; therefore, why did this paper choose a duration of 12 weeks when this period may not be enough to show the full potential of the chosen supplements on memory and cognitive behavior.

3.     How do you ensure that the tablets were taken as directed considering that the study only included two visits (baseline and final)? I think that counting the tablets taken at the end of the study is insufficient since participants may have taken more or less than two tablets per day and not in the time required of them.

4.     To conclude that the effects of a multinutrient formula containing B group vitamins, Ginkgo biloba, and Bacopa monniera supplements are chronic, the biochemical tests must be repeated after 12 weeks and to follow up.

5.     Previous research has shown that there is a link between gender and the effect of multivitamin supplementation; however, it would have been preferable if the cohorts were separated, taking into account that each gender can sometimes necessitate a different duration to illustrate remarkable results. How can you guarantee that the 12 weeks were suitable for both genders?

6.     While Ginkgo Biloba may aid in memory enhancement, it is important to note that it may pose serious health risks, so its use should be reconsidered.

7.     It was stated that "this study chose to focus on shifts in B vitamin levels due to the herbal doses being relatively low compared to other studies." However, it was stated that the aim of this study was to assess the long-term effects of a multinutrient formula containing B group vitamins, Bacopa monniera, and Ginkgo biloba on memory and attention in healthy, middle-aged adults. This being stated, more attention should be focused on the herbal effects.

8.     Please explain and investigate “the treatment-associated benefits on aspects of memory and the possibility that it may be more vulnerable to age-related decline”.

9.     Please revise the introduction and add all needed references.

10.  You must fully introduce the Bacopa monnieri and Ginkgo biloba in the introduction.

Reviewer 3 Report

the article is about the possibility of improving cognition in older age by supplementing with a mixture of vitamins and other memory and attention-enhancing elements such as ginkgo Biloba.   It is  relevant and interesting. It is moderately original, for me the most valuable part of the manuscript is the approach to studying the topic.

The primary value of the text is its design and methodological rigor.  They address the main question posed.

Author Response

The article is about the possibility of improving cognition in older age by supplementing with a mixture of vitamins and other memory and attention-enhancing elements such as ginkgo Biloba.  It is relevant and interesting. It is moderately original; for me the most valuable part of the manuscript is the approach to studying the topic.

The primary value of the text is its design and methodological rigor.  They address the main question posed

Authors: We thank the reviewer for their time and careful consideration of our manuscript.

Round 2

Reviewer 2 Report

Please properly introduce Ginkgo biloba and Bacopa monnieri; there is still a lack of information about them in the introduction.

Author Response

Reviewer 2

Please properly introduce Ginkgo biloba and Bacopa monnieri; there is still a lack of information about them in the introduction.

We have included additional information about Ginkgo biloba and Bacopa monnieri in the introduction that we hope addresses the Reviewer's concerns. Thank you for this recommendation to improve the paper.